# Diffusion-Convolutional Neural Networks

**James Atwood and Don Towsley**
College of Information and Computer Science
University of Massachusetts
Amherst, MA, 01003
{jatwood|towsley}@cs.umass.edu

## Abstract

We present diffusion-convolutional neural networks (DCNNs), a new model for graph-structured data. Through the introduction of a diffusion-convolution operation, we show how diffusion-based representations can be learned from graph-structured data and used as an effective basis for node classification. DCNNs have several attractive qualities, including a latent representation for graphical data that is invariant under isomorphism, as well as polynomial-time prediction and learning that can be represented as tensor operations and efficiently implemented on a GPU. Through several experiments with real structured datasets, we demonstrate that DCNNs are able to outperform probabilistic relational models and kernel-on-graph methods at relational node classification tasks.

## 1   Introduction

Working with structured data is challenging. On one hand, finding the right way to express and exploit structure in data can lead to improvements in predictive performance; on the other, finding such a representation may be difficult, and adding structure to a model can dramatically increase the complexity of prediction

The goal of this work is to design a flexible model for a general class of structured data that offers improvements in predictive performance while avoiding an increase in complexity. To accomplish this, we extend convolutional neural networks (CNNs) to general graph-structured data by introducing a 'diffusion-convolution' operation. Briefly, rather than scanning a 'square' of parameters across a grid-structured input like the standard convolution operation, the diffusion-convolution operation builds a latent representation by scanning a diffusion process across each node in a graph-structured input.

This model is motivated by the idea that a representation that encapsulates graph diffusion can provide a better basis for prediction than a graph itself. Graph diffusion can be represented as a matrix power series, providing a straightforward mechanism for including contextual information about entities that can be computed in polynomial time and efficiently implemented on a GPU.

In this paper, we present diffusion-convolutional neural networks (DCNNs) and explore their performance on various classification tasks on graphical data. Many techniques include structural information in classification tasks, such as probabilistic relational models and kernel methods; DCNNs offer a complementary approach that provides a significant improvement in predictive performance at node classification tasks.

As a model class, DCNNs offer several advantages:

- **Accuracy:** In our experiments, DCNNs significantly outperform alternative methods for node classification tasks and offer comparable performance to baseline methods for graph classification tasks.

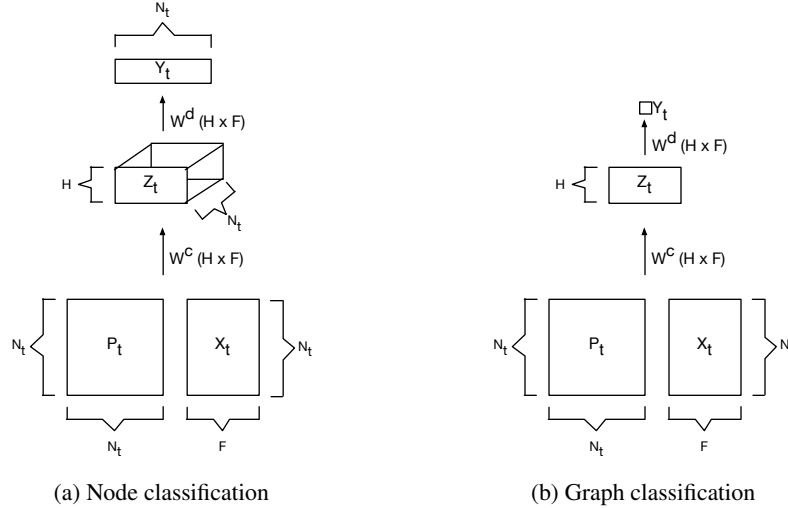

(a) Node classification         (b) Graph classification

Figure 1: DCNN model definition for node and graph classification tasks.

- **Flexibility:** DCNNs provide a flexible representation of graphical data that encodes node features, edge features, and purely structural information with little preprocessing. DCNNs can be used for a variety of classification tasks with graphical data, including node classification and whole-graph classification.

- **Speed:** Prediction from an DCNN can be expressed as a series of polynomial-time tensor operations, allowing the model to be implemented efficiently on a GPU using existing libraries.

The remainder of this paper is organized as follows. In Section 2, we present a formal definition of the model, including descriptions of prediction and learning procedures. This is followed by several experiments in Section 3 that explore the performance of DCNNs at node and graph classification tasks. We briefly describe the limitations of the model in Section 4, then, in Section 5, we present related work and discuss the relationship between DCNNs and other methods. Finally, conclusions and future work are presented in Section 6.

## 2   Model

Consider a situation where we have a set of $T$ graphs $\mathcal{G} = \{G_t | t \in 1...T\}$. Each graph $G_t = (V_t, E_t)$ is composed of vertices $V_t$ and edges $E_t$. The vertices are collectively described by an $N_t \times F$ design matrix $X_t$ of features[1], where $N_t$ is the number of nodes in $G_t$, and the edges $E_t$ are encoded by an $N_t \times N_t$ adjacency matrix $A_t$, from which we can compute a degree-normalized transition matrix $P_t$ that gives the probability of jumping from node $i$ to node $j$ in one step. No constraints are placed on the form of $G_t$; the graph can be weighted or unweighted, directed or undirected. Either the nodes or graphs have labels $Y$ associated with them, with the dimensionality of $Y$ differing in each case.

We are interested in learning to predict $Y$; that is, to predict a label for each of the nodes in each graph or a label for each graph itself. In each case, we have access to some labeled entities (be they nodes or graphs), and our task is predict the values of the remaining unlabeled entities.

This setting can represent several well-studied machine learning tasks. If $T = 1$ (i.e. there is only one input graph) and the labels $Y$ are associated with the nodes, this reduces to the problem of *semisupervised classification*; if there are no edges present in the input graph, this reduces further to standard *supervised classification*. If $T > 1$ and the labels $Y$ are associated with each graph, then this represents the problem of *supervised graph classification*.

DCNNs are designed to perform any task that can be represented within this formulation. An DCNN takes $\mathcal{G}$ as input and returns either a hard prediction for $Y$ or a conditional distribution $\mathbb{P}(Y|X)$. Each

entity of interest (be it a node or a graph) is transformed to a diffusion-convolutional representation, which is a $H \times F$ real matrix defined by $H$ hops of graph diffusion over $F$ features, and it is defined by an $H \times F$ real-valued weight tensor $W^c$ and a nonlinear differentiable function $f$ that computes the activations. So, for node classification tasks, the diffusion-convolutional representation of graph $t$, $Z_t$, will be a $N_t \times H \times F$ tensor, as illustrated in Figure 1a; for graph classification tasks, $Z_t$ will be a $H \times F$ matrix, as illustrated in Figures 1b.

The model is built on the idea of a diffusion kernel, which can be thought of as a measure of the level of connectivity between any two nodes in a graph when considering all paths between them, with longer paths being discounted more than shorter paths. Diffusion kernels provide an effective basis for node classification tasks [1].

The term 'diffusion-convolution' is meant to evoke the ideas of feature learning, parameter tying, and invariance that are characteristic of convolutional neural networks. The core operation of a DCNN is a mapping from nodes and their features to the results of a diffusion process that begins at that node. In contrast with standard CNNs, DCNN parameters are tied according diffusion search depth rather than their position in a grid. The diffusion-convolutional representation is invariant with respect to node index rather than position; in other words, the diffusion-convolututional activations of two isomorphic input graphs will be the same[2]. Unlike standard CNNs, DCNNs have no pooling operation.

**Node Classification**     Consider a node classification task where a label $Y$ is predicted for each input node in a graph. Let $P_t^*$ be an $N_t \times H \times N_t$ tensor containing the power series of $P_t$, defined as follows:

$$P_{tijk}^* = P_{tik}^j \tag{1}$$

The diffusion-convolutional activation $Z_{tijk}$ for node $i$, hop $j$, and feature $k$ of graph $t$ is given by

$$Z_{tijk} = f\left(W_{jk}^c \cdot \sum_{l=1}^{N_t} P_{tijl}^* X_{tlk}\right) \tag{2}$$

The activations can be expressed more concisely using tensor notation as

$$Z_t = f\left(W^c \odot P_t^* X_t\right) \tag{3}$$

where the $\odot$ operator represents element-wise multiplication; see Figure 1a. The model only entails $O(H \times F)$ parameters, making the size of the latent diffusion-convolutional representation independent of the size of the input.

The model is completed by a dense layer that connects $Z$ to $Y$. A hard prediction for $Y$, denoted $\hat{Y}$, can be obtained by taking the maximum activation and a conditional probability distribution $\mathbb{P}(Y|X)$ can be found by applying the softmax function:

$$\hat{Y} = \arg\max\left(f\left(W^d \odot Z\right)\right) \tag{4}$$

$$\mathbb{P}(Y|X) = \text{softmax}\left(f\left(W^d \odot Z\right)\right) \tag{5}$$

This keeps the same form in the following extensions.

**Graph Classification**     DCNNs can be extended to graph classification by taking the mean activation over the nodes

$$Z_t = f\left(W^c \odot 1_{N_t}^T P_t^* X_t / N_t\right) \tag{6}$$

where $1_{N_t}$ is an $N_t \times 1$ vector of ones, as illustrated in Figure 1b.

**Purely Structural DCNNs**     DCNNs can be applied to input graphs with no features by associating a 'bias feature' with value 1.0 with each node. Richer structure can be encoded by adding additional structural node features such as Pagerank or clustering coefficient, although this does introduce some hand-engineering and pre-processing.

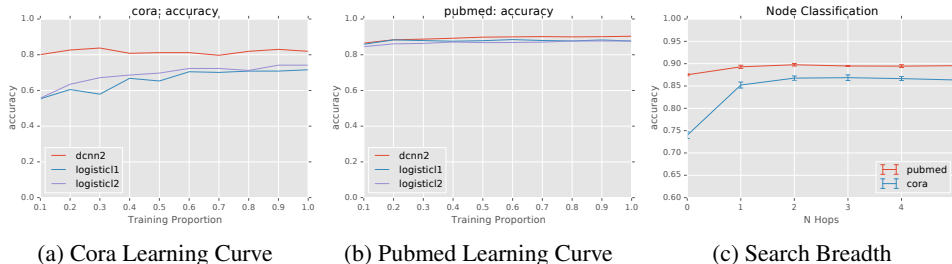

|  | (a) Cora Learning Curve | (b) Pubmed Learning Curve | (c) Search Breadth |

Figure 2: Learning curves (2a - 2b) and effect of search breadth (2c) for the Cora and Pubmed datasets.

**Learning** DCNNs are learned via stochastic minibatch gradient descent on backpropagated error. At each epoch, node indices are randomly grouped into several batches. The error of each batch is computed by taking slices of the graph definition power series and propagating the input forward to predict the output, then setting the weights by gradient ascent on the back-propagated error. We also make use of windowed early stopping; training is ceased if the validation error of a given epoch is greater than the average of the last few epochs.

## 3 Experiments

In this section we present several experiments to investigate how well DCNNs perform at node and graph classification tasks. In each case we compare DCNNs to other well-known and effective approaches to the task.

In each of the following experiments, we use the AdaGrad algorithm [2] for gradient ascent with a learning rate of $0.05$. All weights are initialized by sampling from a normal distribution with mean zero and variance $0.01$. We choose the hyperbolic tangent for the nonlinear differentiable function $f$ and use the multiclass hinge loss between the model predictions and ground truth as the training objective. The model was implemented in Python using Lasagne and Theano [3].

### 3.1 Node classification

We ran several experiments to investigate how well DCNNs classify nodes within a single graph. The graphs were constructed from the Cora and Pubmed datasets, which each consist of scientific papers (nodes), citations between papers (edges), and subjects (labels).

**Protocol** In each experiment, the set $\mathcal{G}$ consists of a single graph $G$. During each trial, the input graph's nodes are randomly partitioned into training, validation, and test sets, with each set having

| | Cora | | | Pubmed | | |
|---|---|---|---|---|---|---|
| Model | Accuracy | F (micro) | F (macro) | Accuracy | F (micro) | F (macro) |
| l1logistic | 0.7087 | 0.7087 | 0.6829 | 0.8718 | 0.8718 | 0.8698 |
| l2logistic | 0.7292 | 0.7292 | 0.7013 | 0.8631 | 0.8631 | 0.8614 |
| KED | 0.8044 | 0.8044 | 0.7928 | 0.8125 | 0.8125 | 0.7978 |
| KLED | 0.8229 | 0.8229 | 0.8117 | 0.8228 | 0.8228 | 0.8086 |
| CRF-LBP | 0.8449 | – | 0.8248 | – | – | – |
| 2-hop DCNN | **0.8677** | **0.8677** | **0.8584** | **0.8976** | **0.8976** | **0.8943** |

Table 1: A comparison of the performance between baseline $\ell 1$ and $\ell 2$-regularized logistic regression models, exponential diffusion and Laplacian exponential diffusion kernel models, loopy belief propagation (LBP) on a partially-observed conditional random field (CRF), and a two-hop DCNN on the Cora and Pubmed datasets. The DCNN offers the best performance according to each measure, and the gain is statistically significant in each case. The CRF-LBP result is quoted from [4], which follows the same experimental protocol.

the same number of nodes. During training, all node features $X$, all edges $E$, and the labels $Y$ of the training and validation sets are visible to the model. We report classification accuracy as well as micro– and macro–averaged F1; each measure is reported as a mean and confidence interval computed from several trials.

We also provide learning curves for the CORA and Pubmed datasets. In this experiment, the validation and test set each contain 10% of the nodes, and the amount of training data is varied between 10% and 100% of the remaining nodes.

**Baseline Methods** 'l1logistic' and 'l2logistic' indicate $\ell 1$ and $\ell 2$-regularized logistic regression, respectively. The inputs to the logistic regression models are the node features alone (e.g. the graph structure is not used) and the regularization parameter is tuned using the validation set. 'KED' and 'KLED' denote the exponential diffusion and Laplacian exponential diffusion kernels-on-graphs, respectively, which have previously been shown to perform well on the Cora dataset [1]. These kernel models take the graph structure as input (e.g. node features are not used) and the validation set is used to determine the kernel hyperparameters. 'CRF-LBP' indicates a partially-observed conditional random field that uses loopy belief propagation for inference. Results for this model are quoted from prior work [4] that uses the same dataset and experimental protocol.

**Node Classification Data** The Cora corpus [5] consists of 2,708 machine learning papers and the 5,429 citation edges that they share. Each paper is assigned a label drawn from seven possible machine learning subjects, and each paper is represented by a bit vector where each feature corresponds to the presence or absence of a term drawn from a dictionary with 1,433 unique entries. We treat the citation network as an undirected graph.

The Pubmed corpus [5] consists of 19,717 scientific papers from the Pubmed database on the subject of diabetes. Each paper is assigned to one of three classes. The citation network that joins the papers consists of 44,338 links, and each paper is represented by a Term Frequency Inverse Document Frequency (TFIDF) vector drawn from a dictionary with 500 terms. As with the CORA corpus, we construct an adjacency-based DCNN that treats the citation network as an undirected graph.

**Results Discussion** Table 1 compares the performance of a two-hop DCNN with several baselines. The DCNN offers the best performance according to different measures including classification accuracy and micro– and macro–averaged F1, and the gain is statistically significant in each case with negligible p-values. For all models except the CRF, we assessed this via a one-tailed two-sample Welch's t-test. The CRF result is quoted from prior work, so we used a one-tailed one-sample test.

Figures 2a and Figure 2b show the learning curves for the Cora and Pubmed datasets. The DCNN generally outperforms the baseline methods on the Cora dataset regardless of the amount of training data available, although the Laplacian exponential diffusion kernel does offer comparable performance when the entire training set is available. Note that the kernel methods were prohibitively slow to run on the Pubmed dataset, so we do not include them in the learning curve.

Finally, the impact of diffusion breadth on performance is shown in Figure 2. Most of the performance is gained as the diffusion breadth grows from zero to three hops, then levels out as the diffusion process converges.

### 3.2 Graph Classification

We also ran experiments to investigate how well DCNNs can learn to label whole graphs.

**Protocol** At the beginning of each trial, input graphs are randomly assigned to training, validation, or test, with each set having the same number of graphs. During the learning phase, the training and validation graphs, their node features, and their labels are made visible; the training set is used to determine the parameters and the validation set to determine hyperparameters. At test time, the test graphs and features are made visible and the graph labels are predicted and compared with ground truth. Table 2 reports the mean accuracy, micro-averaged F1, and macro-averaged F1 over several trials.

We also provide learning curves for the MUTAG (Figure 3a) and ENZYMES (Figure 3b) datasets. In these experiments, validation and test sets each containing 10% of the graphs, and we report the

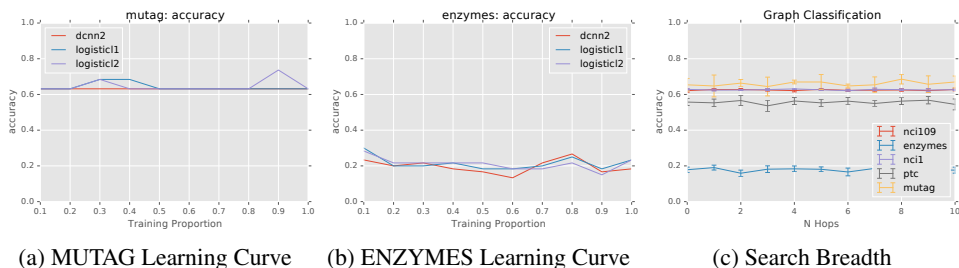

| (a) MUTAG Learning Curve | (b) ENZYMES Learning Curve | (c) Search Breadth |

Figure 3: Learning curves for the MUTAG (3a) and ENZYMES (3b) datasets as well as the effect of search breadth (3c)

performance of each model as a function of the proportion of the remaining graphs that are made available for training.

**Baseline Methods**    As a simple baseline, we apply linear classifiers to the average feature vector of each graph; 'l1logistic' and 'l2logistic' indicate $\ell 1$ and $\ell 2$-regularized logistic regression applied as described. 'deepwl' indicates the Weisfeiler-Lehman (WL) subtree deep graph kernel. Deep graph kernels decompose a graph into substructures, treat those substructures as words in a sentence, and fit a word-embedding model to obtain a vectorization [6].

**Graph Classification Data**    We apply DCNNs to a standard set of graph classification datasets that consists of NCI1, NCI109, MUTAG, PCI, and ENZYMES. The NCI1 and NCI109 [7] datasets consist of 4100 and 4127 graphs that represent chemical compounds. Each graph is labeled with whether it is has the ability to suppress or inhibit the growth of a panel of human tumor cell lines, and each node is assigned one of 37 (for NCI1) or 38 (for NCI109) possible labels. MUTAG [8] contains 188 nitro compounds that are labeled as either aromatic or heteroaromatic with seven node features. PTC [9] contains 344 compounds labeled with whether they are carcinogenic in rats with 19 node features. Finally, ENZYMES [10] is a balanced dataset containing 600 proteins with three node features.

**Results Discussion**    In contrast with the node classification experiments, there is no clear best model choice across the datasets or evaluation measures. In fact, according to Table 2, the only clear choice is the 'deepwl' graph kernel model on the ENZYMES dataset, which significantly outperforms the other methods in terms of accuracy and micro– and macro–averaged F measure. Furthermore, as shown in Figure 3, there is no clear benefit to broadening the search breadth $H$. These results suggest that, while diffusion processes are an effective representation for *nodes*, they do a poor job of summarizing *entire graphs*. It may be possible to improve these results by finding a more effective way to aggregate the node operations than a simple mean, but we leave this as future work.

|  | NCI1 | | | NCI109 | | |
| Model | Accuracy | F (micro) | F (macro) | Accuracy | F (micro) | F (macro) |
| --- | --- | --- | --- | --- | --- | --- |
| l1logistic | 0.5728 | 0.5728 | 0.5711 | 0.5555 | 0.5555 | 0.5411 |
| l2logistic | 0.5688 | 0.5688 | 0.5641 | 0.5586 | 0.5568 | 0.5402 |
| deepwl | 0.6215 | **0.6215** | 0.5821 | 0.5801 | 0.5801 | 0.5178 |
| 2-hop DCNN | 0.6250 | 0.5807 | 0.5807 | 0.6275 | 0.5884 | 0.5884 |
| 5-hop DCNN | **0.6261** | 0.5898 | **0.5898** | **0.6286** | **0.5950** | **0.5899** |

|  | MUTAG | | | PTC | | | ENZYMES | | |
| Model | Accuracy | F (micro) | F (macro) | Accuracy | F (micro) | F (macro) | Accuracy | F (micro) | F (macro) |
| --- | --- | --- | --- | --- | --- | --- | --- | --- | --- |
| l1logistic | **0.7190** | 0.7190 | 0.6405 | 0.5470 | 0.5470 | 0.4272 | 0.1640 | 0.1640 | 0.0904 |
| l2logistic | 0.7016 | 0.7016 | 0.5795 | 0.5565 | **0.5565** | **0.4460** | 0.2030 | 0.2030 | 0.1110 |
| deepwl | 0.6563 | 0.6563 | 0.5942 | 0.5113 | 0.5113 | 0.4444 | **0.2155** | **0.2155** | **0.1431** |
| 2-hop DCNN | 0.6635 | 0.7975 | 0.79747 | **0.5660** | 0.0500 | 0.0531 | 0.1590 | 0.1590 | 0.0809 |
| 5-hop DCNN | 0.6698 | **0.8013** | **0.8013** | 0.5530 | 0.0 | 0.0526 | 0.1810 | 0.1810 | 0.0991 |

Table 2: A comparison of the performance between baseline methods and two and five-hop DCNNs on several graph classification datasets.

# 4   Limitations

**Scalability**   DCNNs are realized as a series of operations on dense tensors. Storing the largest tensor ($P^*$, the transition matrix power series) requires $O(N_t^2 H)$ memory, which can lead to out-of-memory errors on the GPU for very large graphs in practice. As such, DCNNs can be readily applied to graphs of tens to hundreds of thousands of nodes, but not to graphs with millions to billions of nodes.

**Locality**   The model is designed to capture local behavior in graph-structured data. As a consequence of constructing the latent representation from diffusion processes that begin at each node, we may fail to encode useful long-range spatial dependencies between individual nodes or other non-local graph behavior.

# 5   Related Work

In this section we describe existing approaches to the problems of semi-supervised learning, graph classification, and edge classification, and discuss their relationship to DCNNs.

**Other Graph-Based Neural Network Models**   Other researchers have investigated how CNNs can be extended from grid-structured to more general graph-structured data. [11] propose a spatial method with ties to hierarchical clustering, where the layers of the network are defined via a hierarchical partitioning of the node set. In the same paper, the authors propose a spectral method that extends the notion of convolution to graph spectra. Later, [12] applied these techniques to data where a graph is not immediately present but must be inferred. DCNNs, which fall within the spatial category, are distinct from this work because their parameterization makes them transferable; a DCNN learned on one graph can be applied to another. A related branch of work that has focused on extending convolutional neural networks to domains where the structure of the graph itself is of direct interest [13, 14, 15]. For example, [15] construct a deep convolutional model that learns real-valued fingerprint representation of chemical compounds.

**Probabilistic Relational Models**   DCNNs also share strong ties to probabilistic relational models (PRMs), a family of graphical models that are capable of representing distributions over relational data [16]. In contrast to PRMs, DCNNs are deterministic, which allows them to avoid the exponential blowup in learning and inference that hampers PRMs.

Our results suggest that DCNNs outperform partially-observed conditional random fields, the state-of-the-art model probabilistic relational model for semi-supervised learning. Furthermore, DCNNs offer this performance at considerably lower computational cost. Learning the parameters of both DCNNs and partially-observed CRFs involves numerically minimizing a nonconvex objective – the backpropagated error in the case of DCNNs and the negative marginal log-likelihood for CRFs. In practice, the marginal log-likelihood of a partially-observed CRF is computed using a contrast-of-partition-functions approach that requires running loopy belief propagation twice; once on the entire graph and once with the observed labels fixed [17]. This algorithm, and thus each step in the numerical optimization, has exponential time complexity $\mathcal{O}(E_t N_t^{C_t})$ where $C_t$ is the size of the maximal clique in $G_t$ [18]. In contrast, the learning subroutine for an DCNN requires only one forward and backward pass for each instance in the training data. The complexity is dominated by the matrix multiplication between the graph definition matrix $A$ and the design matrix $V$, giving an overall polynomial complexity of $\mathcal{O}(N_t^2 F)$.

**Kernel Methods**   Kernel methods define similarity measures either between nodes (so-called kernels on graphs) [1] or between graphs (graph kernels) and these similarities can serve as a basis for prediction via the kernel trick. The performance of graph kernels can be improved by decomposing a graph into substructures, treating those substructures as a words in a sentence, and fitting a word-embedding model to obtain a vectorization [6].

DCNNs share ties with the exponential diffusion family of kernels on graphs. The exponential diffusion graph kernel $K_{ED}$ is a sum of a matrix power series:

$$K_{ED} = \sum_{j=0}^{\infty} \frac{\alpha^j A^j}{j!} = \exp(\alpha A) \tag{7}$$

The diffusion-convolution activation given in (3) is also constructed from a power series. However, the representations have several important differences. First, the weights in (3) are learned via backpropagation, whereas the kernel representation is not learned from data. Second, the diffusion-convolutional representation is built from both node features and the graph structure, whereas the exponential diffusion kernel is built from the graph structure alone. Finally, the representations have different dimensions: $K_{ED}$ is an $N_t \times N_t$ kernel matrix, whereas $Z_t$ is a $N_t \times H \times F$ tensor that does not conform to the definition of a kernel.

# 6 Conclusion and Future Work

By learning a representation that encapsulates the results of graph diffusion, diffusion-convolutional neural networks offer performance improvements over probabilistic relational models and kernel methods at node classification tasks. We intend to investigate methods for a) improving DCNN performance at graph classification tasks and b) making the model scalable in future work.

# 7 Appendix: Representation Invariance for Isomorphic Graphs

If two graphs $G_1$ and $G_2$ are isomorphic, then their diffusion-convolutional activations are the same. Proof by contradiction; assume that $G_1$ and $G_2$ are isomorphic and that their diffusion-convolutional activations are different. The diffusion-convolutional activations can be written as

$$Z_{1jk} = f\left(W_{jk}^c \odot \sum_{v \in V_1} \sum_{v' \in V_1} P_{1vjv'}^* X_{1v'k} / N_1\right)$$

$$Z_{2jk} = f\left(W_{jk}^c \odot \sum_{v \in V_2} \sum_{v' \in V_2} P_{2vjv'}^* X_{2v'k} / N_2\right)$$

Note that

$$V_1 = V_2 = V$$
$$X_{1vk} = X_{2vk} = X_{vk} \ \forall \ v \in V, k \in [1, F]$$
$$P_{1vjv'}^* = P_{2vjv'}^* = P_{vjv'}^* \ \forall \ v, v' \in V, j \in [0, H]$$
$$N_1 = N_2 = N$$

by isomorphism, allowing us to rewrite the activations as

$$Z_{1jk} = f\left(W_{jk}^c \odot \sum_{v \in V} \sum_{v' \in V} P_{vjv'}^* X_{v'k} / N\right)$$

$$Z_{2jk} = f\left(W_{jk}^c \odot \sum_{v \in V} \sum_{v' \in V} P_{vjv'}^* X_{v'k} / N\right)$$

This implies that $Z_1 = Z_2$ which presents a contradiction and completes the proof.

**Acknowledgments**

We would like to thank Bruno Ribeiro, Pinar Yanardag, and David Belanger for their feedback on drafts of this paper. This work was supported in part by Army Research Office Contract W911NF-12-1-0385 and ARL Cooperative Agreement W911NF-09-2-0053. This work was also supported by NVIDIA through the donation of equipment used to perform experiments.

## Footnotes

[1]Without loss of generality, we assume that the features are real-valued.

[2]A proof is given in the appendix.

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
