[Reviews · NeurIPS 2016]

Reviewer 1

Summary

This paper combines multiple graph kernels with a diffusion-convolutional neural net. Specifically, they propose weighting the jth hop transition for feature k in the first layer and then outputting an activation for each node, hop, feature and graph. This way, the weights can be learned from the data.

Qualitative Assessment

The paper is of significant importance, since the problem of learning kernels from graph labels has not been comprehensively addressed in the literature. The paper could be much clearer, since the power spectrum P star and the hop is not well defined. I think that I know what is going on, but the average reader may not. The experiments are very informative and comprehensive.

Confidence in this Review

2-Confident (read it all; understood it all reasonably well)


Reviewer 2

Summary

This paper proposes a neural network model for node, edge and graph classification problems that is based on integrating information of the local vicinity of each node in the graph. They obtain good results on node classification but not on graph classification.

Qualitative Assessment

This paper presents what seems to be a reasonable model for node classification that uses local information to improve performance. However, as someone who is not an expert in this field is was hard for me to assess the quality for several reasons: 1. The authors get good results on node classification but not on graph classification. However, even for the node classification results, it seemed most of the baselines are not exposed to the same information as the method so it is hard to tell whether difference in performance is due to that and not to the model. 2. The authors mention in related work that there are other works on using neural network models for graphs, but they do not compare to those anywhere, why? 3. The authors define models for edge classification but there are no empirical results. why? To conclude, the authors present a reasonable model for using neural networks for graphs, but as someone who is not an expert it was hard to understand the extent of the contribution compared to existing literature. Minor: Figure 1 is too small

Confidence in this Review

2-Confident (read it all; understood it all reasonably well)


Reviewer 3

Summary

This paper extends CNNs to handle general graph-structured data by introducing a diffusion-convolution operation, extensive experiments are conducted on various node and graph classification tasks, and better performance results are achieved over probabilistic relational models and kernel methods.

Qualitative Assessment

lines 109 - 110 "training is ceased if the validation error of a given epoch is greater than the average of the last few epochs." Do you try other criteria, such as if the validation error of a given epoch is greater than the last epoch, cut learning rate in half? line 98 "graph" showed twice.

Confidence in this Review

2-Confident (read it all; understood it all reasonably well)


Reviewer 4

Summary

The authors propose a new convolutional neural network (CNN) based model called the diffusion convolutional network (DCNN) for classifying graph-structured data at the level of nodes, edges, and entire graphs. The fundamental operation in a DCNN is a mapping from nodes and their features to the results of a diffusion process starting at those nodes. Unlike standard CNNs, DCNN parameters are tied according to search depth rather than their position in a grid. The authors establish theoretically that the DCNN activations of two isomorphic input graphs are the same, ie, the DCNN representation is invariant with respect to node index. The authors compare the DCNN model with other models on node and graph classification tasks, and report state of the art performance on the former and comparable performance on the latter.

Qualitative Assessment

The authors propose a novel model called diffusion convolutional network (DCNN) for classifying graph-structured data at the level of nodes, edges, and entire graphs. The model is impressive in that it achieves state of the art performance on node classification tasks. But it doesn't do as well on graph classification tasks. The authors do not present any results of applying the DCNN model on edge classification tasks. The authors have done a good job explaining their model, but they could have been clearer with the notation and figures 1 (a-c) by using a small graph as a running example.

Confidence in this Review

2-Confident (read it all; understood it all reasonably well)


Reviewer 5

Summary

The paper proposed a new neural network model for classification tasks over generic graphs, as an extension of CNN. The model explicitly takes into account graph diffusion via a multi-hop architecture, and is able to leverage both node attributes and the graph structure during training. Although the model outperforms several simple baselines on average, it seems to suffer from relatively poor scalability.

Qualitative Assessment

The idea of incorporating graph diffusion into neural networks seem both interesting and novel. The authors also did a good job in motivating the problem. However, overall I feel several aspects of the work could be further improved: Scalability: 1. The authors proposed three separate models for node, graph and edge classification. However, no empirical performance of edge classification was reported. I was wondering if this is due to scalability issues as for edge classification the parameter size can be extremely large even for medium-scale graphs (according to eq (6)). 2. It would be helpful to include the run time of the current experiments. About the experiment settings: 1. Experiment comparisons seem unfair. While DCNN uses both node features and graph structures, at least 4 out of the 5 baselines (l1-logistic, l2-logistic, KED, KLED) are solely relying on graph structures as their input, making it inconclusive whether the performance gain of DCNN is due to a good model design or simply because of the additional side information. To make the comparison more meaningful, at least the performance of the purely structural DCNN (101-104) should be reported. 2. Section 5 mentioned about several recent neural architectures for graph-based learning based similar idea of extending CNN to irregular graph domains. However, none of those models were included as baselines for empirical comparison. Without strong baselines, it is hard to judge whether the proposed DCNN is actually achieving the state-of-the-art performance (or close). Relationship to CNN: Unlike existing works that explicitly generalizes convolution operator over 2d-grid [1,2], it is not clear whether DCNN proposed in this work can be viewed as an extension of CNN(as claimed by the authors), since it is not obvious how DCNN will subsume CNN as a special case. [1] Bruna, Joan, et al. "Spectral networks and locally connected networks on graphs." arXiv preprint arXiv:1312.6203 (2013). [2] Henaff, Mikael, Joan Bruna, and Yann LeCun. "Deep convolutional networks on graph-structured data." arXiv preprint arXiv:1506.05163 (2015).

Confidence in this Review

2-Confident (read it all; understood it all reasonably well)


Reviewer 6

Summary

The authors propose a novel neural network architecture for prediction problems of graphs. They interpret this as a generalization of convolution beyond grid structures. The method is shown to outperform several kernel methods. It performs better on vertex classification, but not on graph classification.

Qualitative Assessment

The experimental results look promising, but I have several concerns regarding the presentation and the experiments. Presentation: - How exactly is diffusion defined here? This is such a core concept, why is there not a formal definition or at least a citation to previous work? I spent 10 minutes googling this, and I haven't found a single obvious definition. I'm not an expert on graphs / social network analysis, but this still seems to be an important omission. In any case, I assume there is some standard way to featurize a node's neighbors that is used here. - Authors strongly emphasize how this work generalizes convolutions. I think this is only true in the graph setting, not in the edge or vertex prediction settings (e.g. I don't see where is the spatial invariance in those settings). Since the contribution is better performance on vertex classification, I don't think it's appropriate to emphasize this story. Experiments: - One of the claims is that this architecture handles vertex, edge, graph prediction in a unified way. Why are there no edge experiments? There should be at least a comment on this. Also, can it be used for structure learning? - On vertex prediction, the authors compare against baselines including kernel methods, logistic regression and a CRF. But logistic regression only receives features derived from the vertex itself, not its neighbors, so it's not surprising they don't do well. There needs to be experiments that trains a shallow model on the same signal (i.e. on features that are comparable to the diffusion-based features used here). Then we need to see that the neural network model is more accurate or faster. The same holds for the CRF; the authors don't specify which features it uses, but even if it uses edge features, we need to compare against an equivalent featurization. The authors mention they use up to 5 hops of diffusion, which sounds to me like they're looking at neighbors 5 hops away. The CRF and LR need to receive the same features. - I'm not an expert on neural networks applied to graphs, but I'm surprised there are not other neural-network based models to compare against. Why is something like this work not applicable and/or compared against: https://cs.stanford.edu/people/jure/pubs/node2vec-kdd16.pdf In conclusion, I find the architecture to be elegant and the paper very well-written (except for omitting the definition of diffusion), but the experiments are not entirely convincing, and this is the main factor by which we can judge an applied paper. Right now, I think this paper is somewhat borderline, but I am happy to readjust my opinion if the authors can comment on some of the above. UPDATE: After reading the response and looking closer to related literature, I am down-grading my review. I strongly disagree with the authors' claim that "many of [other] techniques were developed in parallel with our own and did not have readily available open-source implementations". For example, look at the node2vec paper that came out in parallel: - They have substantially more extensive experiments - They compare against previous neural network based methods, LINE and DeepWalk - These methods are from 2014 and 2015, so I see no excuse for not comparing against them. In fact, they are not even cited, which I think is unacceptable! The quality of this paper is much lower than that of LINE, DeepWalk and node2vec, and so I strongly argue for the rejection of this paper.

Confidence in this Review

2-Confident (read it all; understood it all reasonably well)